# Risperidone Exacerbates Glucose Intolerance, Nonalcoholic Fatty Liver Disease, and Renal Impairment in Obese Mice

**DOI:** 10.3390/ijms22010409

**Published:** 2021-01-02

**Authors:** Hsiao-Pei Tsai, Po-Hsun Hou, Frank-Chiahung Mao, Chia-Chia Chang, Wei-Cheng Yang, Ching-Feng Wu, Huei-Jyuan Liao, Tzu-Chun Lin, Lan-Szu Chou, Li-Wei Hsiao, Geng-Ruei Chang

**Affiliations:** 1Ph.D. Program of Agriculture Science, National Chiayi University, 300 Syuefu Road, Chiayi 60004, Taiwan; tsaibelle@mail.ncyu.edu.tw; 2Veterinary Teaching Hospital, National Chiayi University, 580 Xinmin Road, Chiayi 60054, Taiwan; 3Department of Psychiatry, Taichung Veterans General Hospital, 4 Section, 1650 Taiwan Boulevard, Taichung 40705, Taiwan; peterhopo2@yahoo.com.tw; 4Faculty of Medicine, National Yang-Ming University, 2 Section, 155 Linong Street, Beitou District, Taipei 11221, Taiwan; 5Department of Veterinary Medicine, National Chung Hsing University, 250 Kuo Kuang Road, Taichung 40227, Taiwan; fcmao@nchu.edu.tw; 6Animal Drugs Inspection Branch, Animal Health Research Institute, Council of Agriculture, 21 Muchang, Ciding Village, Zhunan Township, Miaoli County 35054, Taiwan; bccchia@mail.nvri.gov.tw; 7School of Veterinary Medicine, National Taiwan University, 4 Section, 1 Roosevelt Road, Taipei 10617, Taiwan; yangweicheng@ntu.edu.tw; 8Division of Thoracic and Cardiovascular Surgery, Department of Surgery, Chang Gung University, Chang Gung Memorial Hospital, Linkou, 5 Fuxing Street, Guishan District, Taoyuan 33305, Taiwan; maple.bt88@gmail.com (C.-F.W.); lin890090@gmail.com (T.-C.L.); 9Department of Veterinary Medicine, National Chiayi University, 580 Xinmin Road, Chiayi 60054, Taiwan; pipi324615@gmail.com; 10Department of BioAgricultural Sciences, National Chiayi University, 300 Syuefu Road, Chiayi 60004, Taiwan; 11Division of Endocrinology and Metabolism, Chang Bing Show Chwan Memorial Hospital, 6 Lugong Road, Changhua 50544, Taiwan

**Keywords:** fatty liver disease, glucose intolerance, obesity, renal impairment, risperidone

## Abstract

Risperidone, a second-generation antipsychotic drug used for schizophrenia treatment with less-severe side effects, has recently been applied in major depressive disorder treatment. The mechanism underlying risperidone-associated metabolic disturbances and liver and renal adverse effects warrants further exploration. This research explores how risperidone influences weight, glucose homeostasis, fatty liver scores, liver damage, and renal impairment in high-fat diet (HFD)-administered C57BL6/J mice. Compared with HFD control mice, risperidone-treated obese mice exhibited increases in body, liver, kidney, and retroperitoneal and epididymal fat pad weights, daily food efficiency, serum triglyceride, blood urea nitrogen, creatinine, hepatic triglyceride, and aspartate aminotransferase, and alanine aminotransferase levels, and hepatic fatty acid regulation marker expression. They also exhibited increased insulin resistance and glucose intolerance but decreased serum insulin levels, Akt phosphorylation, and glucose transporter 4 expression. Moreover, their fatty liver score and liver damage demonstrated considerable increases, corresponding to increases in sterol regulatory element-binding protein 1 mRNA, fatty acid-binding protein 4 mRNA, and patatin-like phospholipid domain containing protein 3 expression. Finally, these mice demonstrated renal impairment, associated with decreases in glutathione peroxidase, superoxide dismutase, and catalase levels. In conclusion, long-term administration of risperidone may exacerbate diabetes syndrome, nonalcoholic fatty liver disease, and kidney injury.

## 1. Introduction

Risperidone is a second-generation antipsychotic drug derived from benzisoxazole that is used in the treatment of various psychiatric conditions, including sleep disturbances, sexual dysfunction, bipolar disorder, schizophrenia, depression, autism, and attention deficit hyperactivity disorder [1]. As such, it is chemically distinct from olanzapine and clozapine, which are dibenzodiazepines and atypical antipsychotic agents. Risperidone is usually effective in animal behavioral models and is considered predictive of antipsychotic activity, such as the suppression of apomorphine- and amphetamine-induced stereotypy and conditioned avoidance behavior [2]. Moreover, risperidone was demonstrated to exert strong anti-aggressive effects in an animal model of escalated aggression during immature stages of development, which may mean that its specific effects on this subtype of aggression could be reflective of its effects on adolescents and children [2]. Its mechanism of action remains unclear, although it may be an antagonist related to serotonin and dopamine receptor inhibition [3,4]. Risperidone can increase extracellular concentrations of serotonin and dopamine turnover by blocking serotonin receptors [3]. This benefit produces marked effects on serotonin, and a strong serotonin antagonist, such as risperidone, may counteract the most lethal and serious side effects of antidepressants, such as serotonin syndrome (the mortality rate is ≤11%), particularly in combination with serotonin reuptake inhibitors for the treatment of mental disorders [4].

The primary antipsychotics adopted in clinics are second-generation antipsychotics, and they offer the main advantage of a favorable side-effect profile; in terms of extrapyramidal symptoms in particular, risperidone has few to none. Moreover, it has been proposed to offer more substantial relief for unpleasant symptoms than first-generation antipsychotics when administered at a clinically effective dose [5]. Some reports have indicated that patients taking second-generation antipsychotics may exhibit cardiovascular risk factors, uncontrollable weight gain, obesity, dyslipidemia, diabetes, or hyperglycemia [6,7]. Such negative effects could make patients less likely to comply with medical instructions and lead to a rise in health costs, potentially imposing substantial financial and physical strain on patients and their relatives. However, compared with treatment with other second-generation antipsychotic drugs (e.g., haloperidol, clozapine, and olanzapine), long-term administration of risperidone has a lower risk of altering physical parameters, such as body weight and lipid profile (including plasma triglyceride level) [8,9,10]. A case report of hypoglycemia was excluded in one study because the patient had noninsulin-dependent diabetes mellitus and was taking risperidone concurrently with glyburide and sertraline [11]. Moreover, some studies have reported that risperidone induces hypoglycemia by increasing insulin secretion in nondiabetic patients with schizophrenia [12,13]. One of these reports indicated that a reduction in risperidone dose mitigated its effect of induced hypoglycemia. Furthermore, a study reported that switching from olanzapine to risperidone for the treatment of schizophrenia exacerbated a patient’s diabetes mellitus and hyperglycemia and resulted in decreased blood glucose levels and increased insulin levels [14]. Evidence suggests an association between risperidone use and a reduced risk of abnormalities during glucose–insulin homeostasis [13,14]. However, some reports have indicated that risperidone is associated with hyperglycemia and exacerbates preexisting diabetes and hypertriglyceridemia in patients on antipsychotic therapy [15,16,17] and that its long-term use induces nonalcoholic fatty liver disease-associated visceral adiposity [18] as well as liver damage [19] in animal models. A recent retrospective cohort study involved a large database analysis that raised concerns regarding a potential risk of kidney injury (particularly interstitial nephritis, glomerulonephritis, and chronic kidney disease (CKD)) associated with antipsychotics, including risperidone, aripiprazole, haloperidol, fluphenazine, olanzapine, and ziprasidone [20,21].

Obesity and negative metabolic effects (in particular, endocrine and metabolic defects, and rapidly increasing weight) are major concerns for patients taking such antipsychotics [22]. The apparent cause of dyslipidemia—a feature of insulin resistance (IR) syndrome, characterized by elevated triglyceride and small dense low-density lipoprotein levels with low high-density lipoprotein levels—is insulin activity inhibition [8,23]. Triglycerides have been proposed to potentially be a major factor in IR pathogenesis—the main type 2 diabetes (T2D) predictor [24]. Moreover, hyperinsulinemia and glucose intolerance, which are commonly observed with IR, may cause intrahepatic triglyceride overproduction in patients with nonalcoholic fatty liver disease [25,26]. Studies have estimated that patients with medication-treated schizophrenia have an approximately threefold higher risk of metabolic syndrome compared with community and clinical samples from the general population and that the prevalence of metabolic syndrome among this population, including IR, rapid weight increase, diabetes, elevated cardiovascular risk, and dyslipidemia, is 14.7–68% [27,28]. Furthermore, recent clinical studies have revealed that patients with schizophrenia have a greater likelihood of developing liver disease, including alcohol-related cirrhosis, chronic liver disease, and nonalcoholic fatty liver disease [29,30]. It has been postulated that this polymorphism makes patients more susceptible to severe increases in triglycerides following antipsychotic treatment, consequently leading to liver injury. Furthermore, the heightened risk of developing CKD among patients with incident schizophrenia may be associated with second-generation antipsychotic–induced metabolic syndrome [31]. The increase in risk of CKD varies from small to moderate among patients taking second-generation antipsychotics, and whether individual atypical and typical antipsychotics are associated with differential kidney damage risks has not been fully investigated in patients with obesity who have psychotic disorders [21,31].

Consistent with this aforementioned notion, the observed differential effects of risperidone are probably the result of discrepancies among metabolic syndrome, slight metabolic syndrome, and non–metabolic syndrome in terms of changes in body weight and hyperglycemia attributed to risperidone. Therefore, we investigated high-fat diet (HFD)–administered mice treated with risperidone to mimic obesity treatment to explore the potential effects of risperidone in obese patients with psychotic disorders. Furthermore, the focus of most studies exploring risperidone’s metabolic side effects has been on changes in glucose levels and lipid metabolism syndrome, and risperidone’s influence on fatty liver and kidney damage is little understood. Our research also sought clarification regarding risperidone’s metabolic effects on hepatic and renal function in obese animals. These results provide further important insight into the metabolic effects and fatty liver and renal damage mechanisms of long-term use of risperidone as a psychotropic agent in patients, and clarify whether the drug worsens metabolic abnormalities, and liver and kidney injury.

## 2. Results

### 2.1. Risperidone Affects the Food Efficiency and Intake and Morphometric Parameters

The preliminary research we conducted did not reveal any significantly different anti-obesity effects among mice receiving a standard diet (SD) between those with and those without risperidone treatment (Appendix A). However, surprisingly, HFD-fed mice who were treated with risperidone for 8 weeks exhibited elevations in morphometric parameters and serum leptin levels compared with the saline controls (Figure 1). Significant increases in weight (Figure 1a) were observed following treatment with risperidone. The risperidone-treated mouse body weight was 1.2-fold higher (Figure 1a) than those of controls (HFD-fed mice; Figure 1a). Furthermore, the weekly food intake increased significantly by 22% in mice receiving risperidone, which paralleled the increase in food intake among controls (Figure 1b). These changes occurred along with a significant increase in daily food efficiency among mice treated with risperidone that was 38% higher than that in the control mice (Figure 1c). Leptin as a nutritional signal, regulating appetite and elevated leptin concentrations, are associated with loss of appetite [32]. Here, compared with the controls, HFD-fed mice who received risperidone treatment exhibited a significant 1.4-fold increase in serum leptin levels (Figure 1d), in contrast to the data on food intake.

### 2.2. Risperidone Increases the Weight of Organs and Fat Pads

We performed assessments to determine whether the observed weight differences were related to changes in adiposity or body composition. At 8 weeks after commencing treatment with risperidone on a HFD diet, the mice had significantly different body compositions compared with the control mice in terms of their epididymal white adipose tissue (EWAT), retroperitoneal white adipose tissue (RWAT), kidney, and liver, but not in terms of their hearts or spleens (Figure 2). Moreover, the treated mice had significantly higher liver, kidney, RWAT, and EWAT weights (by 27%, 16%, 51%, and 20%, respectively) compared with control mice after body weight normalization. The weights of the kidneys and heart as a proportion of total body weight did not differ, however, between the risperidone-treated mice and control group.

### 2.3. Risperidone Restricts the Accumulation of Liver Fat and Size of Adipocytes

Morphometric analysis based on hematoxylin and eosin (H&E) staining of the control and risperidone-treated mouse tissues indicated that, in comparison with the control mice, the treated mice had marked increases in RWAT and EWAT adipocyte sizes and liver fat levels (Figure 3a), suggesting that risperidone blocks the accumulation of fat in the liver, thus increasing hypertrophy of the fat pads. Our analysis of changes in fatty liver scores (Figure 3b) and in adipocyte size of RWAT (Figure 3c) and EWAT (Figure 3d) suggested significantly different results between the groups, namely in that the treated mice had nearly 1.5-fold higher fatty liver scores than did the control mice. Moreover, the mean adipocyte size of RWAT and EWAT was significantly greater (15% and 18%, respectively) in the treatment group, which accords with the higher fat pad weights observed in the treated mice. That is, the risperidone-treated mice had fewer RWAT and EWAT adipocytes of a 0–50-μm and 50–100-μm diameter but more of those of a 100–150-μm and >150 μm diameter—as indicated in Table 1. Therefore, even though a marked increasing tendency in RWAT and EWAT adipocyte size was observed in response to an HFD, risperidone accelerated this increase in adipocyte size.

### 2.4. Risperidone Increases the Serum Levels of Alanine Aminotransferase, Aspartate Aminotransferase, Fatty Acid-Binding Protein 4 mRNA, and Sterol Regulatory Element-Binding Protein 1 mRNA

In the treated mice, serum alanine aminotransferase (ALT) levels increased by 26% (Figure 4a), while serum aspartate aminotransferase (AST) levels increased by 20% (Figure 4b)—constituting a hepatic function marker. Implicated in the regulation of numerous metabolic pathways including T2D, hepatic lipid accumulation, and atherosclerosis [23], fatty acid-binding protein 4 (FABP4), and sterol regulatory element-binding protein 1 (SREBP1) have potential dominant roles in fatty liver disease promotion in rats and humans [33]. In the present study, risperidone-treated mice exhibited significant 3.8-fold (Figure 4c) and 2-fold (Figure 4d) increases, respectively, in mRNA levels of *FABP4* and *SREBP1* compared with the control mice. Genetic activation of hepatic steatosis caused substantial increases in ALT and AST levels, possibly because the contribution of risperidone to fatty liver disease development was mediated by increased mRNA expression of molecular mechanisms of lipid accumulation in the liver.

### 2.5. Risperidone Increases the Serum and Hepatic Levels of Triglycerides as Well as the Hepatic Expression of Patatin-Like Phospholipid Domain Containing Protein 3 and Fatty Acid Synthase but Reduces the Hepatic Expression of Adiponectin and AMPK

The serum and hepatic triglyceride levels were 27% (Figure 5a) and 28% (Figure 5b) in the treated mice than in the control mice. Fatty acid synthase (FASN) is a crucial liver enzyme for lipid homeostasis and triglyceride synthesis [23], whereas patatin-like phospholipid domain containing protein 3 (PNPLA3) is a pathologic marker for lipogenesis regulation in obesity, nonalcoholic fatty liver disease, and cardiovascular disease [25]. Western blot analysis (Figure 5c) indicated that hepatic FASN and PNPLA3 expression was 42% (Figure 5d) and 17% (Figure 5e) higher in the treated mice than in the control mice. Thus, risperidone upregulated lipogenesis in the liver—thus increasing HFD-induced fatty liver scores.

### 2.6. Risperidone Reduces Insulin Levels and Worsens Glucose Intolerance

The glucose tolerance of SD-fed mice was evaluated after 7 weeks of treatment with risperidone. Significant risperidone-induced impairment in glucose tolerance was observed in comparison with the untreated mice (Figure 6a); this result accords with the observation that the risperidone-treated mice had higher body weight gain and increased fatty liver scores, which both typically reduce insulin sensitivity (IS) [34]. As expected, risperidone-treated mice exhibited significantly increased fasting blood glucose levels at 30, 60, 90, and 120 min after injection. Relative to baseline levels, 20% and 9% increases in such glucose levels were observed at 120 min after injection in the treated and control mice, respectively. Moreover, the glucose levels of risperidone-treated mice had a significantly (1.1-fold) higher area under the curve (AUC) at 120 min (Figure 6b). A significant proportion of the mice receiving risperidone treatment exhibited glucose intolerance, which was defined in this study as >9 mmol/L blood glucose at 120 min postinjection (Figure 6c). Furthermore, we observed significantly lower serum insulin levels in treated mice that those in control mice (Figure 6d). Therefore, risperidone could ameliorate diabetes symptoms, including hyperglycemia exacerbation and hypoinsulinemia-associated glucose tolerance impairment in obese mice.

### 2.7. Risperidone Reduces IS by Attenuating Phosphorylated Akt and Glucose Transporter 4 Expression

To assess IR and IS, we used the homeostatic model assessment for IR (HOMA-IR) and IS indices, respectively [25,35,36]—which demonstrated 71% increase and 36% decreased in the treated mice compared with the control mice, respectively (Figure 7a and Figure 7b, respectively). Evaluation of the effects of risperidone on insulin signaling in muscles indicated reduced Akt activation and glucose transporter 4 (GLUT4) expression after risperidone treatment (Figure 7c). Muscle Akt activation and GLUT4 expression decreased by 17% and 54% in the treatment group compared with those in the control group, respectively (Figure 7d,e, respectively).

### 2.8. Risperidone Induces Renal Injury and Increases Serum Blood Urea Nitrogen and Creatinine Levels but Reduces Antioxidant Enzymes in the Kidneys

Next, because obesity, hyperlipidemia, and diabetes lead to the development of renal injury, we evaluated whether risperidone induces renal damage [37]. H&E staining revealed that risperidone treatment in mice induced glomerulonephritis with inflammatory cell infiltration with the control mice. Moreover, analysis results indicated that serum levels of creatinine and blood urea nitrogen (BUN) were significantly increased (by 2.2- and 2.1-fold, respectively) in risperidone-treated mice compared with control mice (Figure 8b and Figure 8c, respectively). The development of renal damage is linked to the reduction of antioxidant enzymes in the kidney, and this deceased antioxidant activity may cause necrosis and impaired renal function [38]. Indeed, mice exhibited significant 16%, 35%, and 41% decreases in the antioxidant biomarkers catalase (Figure 8d), glutathione peroxidase (GPx; Figure 8e), and superoxide dismutase (SOD; Figure 8f), respectively, after risperidone treatment compared with the control mice.

## 3. Discussion

This research explored how risperidone influences the development of obesity in C57BL/6J mice on an HFD diet. Our findings indicated that mice receiving a 56-day risperidone treatment had a higher rate of obesity, with increases in visceral fat and the development of fatty liver disease, glucose intolerance, and IR. Daily food efficiencies, liver weights, kidney weights, fat pad weights, adipocyte sizes, fatty liver scores, and serum and hepatic triglyceride levels had also increased. We could thus verify risperidone’s ability to not only increase food intake and body weight gain but also impair homeostasis of glucose with hypoinsulinemia. In addition to increases in fatty liver scores, risperidone has been proposed to be linked to adipogenesis in the liver caused by associated proteins, including FASN and PNPLA3, as well as phosphorylation activation in risperidone-treated mice. Changes in IR typically exhibit an association with lower insulin signaling protein activity, and our risperidone-treated mice with obesity and hyperglycemia exhibited impaired glucose homeostasis due to the reduction of Akt phosphorylation and GLUT4 expression. After 56 days of risperidone treatment, mice had a significantly increased kidney function index and decreased antioxidant enzymes compared with the control mice, corresponding to eventual renal injury.

An 8-week HFD contributed to the development of obesity in mice, but risperidone generated increases in body, kidney, liver, RWAT, and EWAT weights. This increase in body weight is linked to the increase in fat pad weight due to increased adipocyte differentiation (precursor cells’ delay in generating adipocytes) or adipocyte hypertrophy (shrinking adipocytes as a result of fat storage) [22,32]. Therefore, the increase in body fat among mice in the treatment group could be attributable to growths in mean adipocyte size and fatty liver score. Moreover, the increase in number of large adipocytes due to risperidone treatment may be attributable to the ability of chronic risperidone treatment to suppress energy expenditure through reductions in total resting metabolic rate and nonaerobic metabolism in mice [39]. Another explanation for this result is risperidone’s promotion of fat adipogenesis, which accords with reported findings that risperidone can increase or activate adipogenic marker gene expression—such as through protein kinases A and C-β, tumor growth factor-β, extracellular signal-related kinase (ERK) 1/2, or hormone-sensitive lipase expression—which in turn influences lipogenesis and HFD-induced adipocyte hypertrophy [25,40,41,42]. Moreover, the synthesis of leptin, a food intake regulation hormone produced in hypothalamus, predominantly occurs in white adipose tissue (WAT) [24]. Increasing food intake can usually lead to increases in body weight. Mice receiving risperidone treatment had a higher food intake but higher serum leptin levels. The leptin signaling pathway was inhibited after risperidone treatment, providing support for a report that leptin signaling is inhibited by risperidone through its modulation of suppressor of cytokine signaling (SOCS) 3 and SOCS 6 expression as a result of adenylate cyclase–mediated ERK activation in vitro [42]. Furthermore, we discovered that leptin receptor levels were decreased in mice receiving risperidone treatment (Appendix A). The possibility that long-term risperidone use leads to leptin resistance with the inhibition of leptin signaling is a valuable finding.

Most studies on HFD-induced obesity in mice have indicated that in adipose tissues, serotonin can regulate de novo lipogenesis and systemic energy homeostasis through serotonin 2A receptor signaling [43,44]. The use of medication to inhibit serotonin synthesis causes lipogenesis to be inhibited in EWAT and adaptive thermogenesis to be activated in in brown adipose tissue. A conflicting report demonstrated an association of using a selective serotonin transporter inhibitor to enhance serotonin activity with weight loss; however, the effects were short lived, with a return to the original weight during the maintenance period [45]. Such conflicting findings were also observed in our research. In general, the aforementioned results corroborate our findings, particularly pertaining to large increases in the weights of WAT and liver, levels of serum triglycerides and hepatic triglycerides, and fatty liver scores as a result of elevated serum serotonin levels in mice on an HFD receiving risperidone treatment (Appendix A). Evidence supports that higher serotonin levels can promote lipid accumulation and hepatic steatosis development in mice [43,46]. All these results collectively suggest that serotonin signaling inhibition is an effective strategy for obesity treatment in patients with psychotic disorders who are taking risperidone.

Fat accumulation in the liver may lead to fatty liver disease and subsequently cause elevations in liver enzymes and hepatic injury markers, such as ALT and AST [25,47]. The degree of liver involvement ranges from simple steatosis to steatohepatitis and finally to cryptogenic cirrhosis [47]. Indeed, risperidone-treated mice had higher fatty liver scores, along with higher levels of ALT and AST. We also analyzed *FABP4* and *SREBP1* mRNA expression, which were linked to the expression of genes responsible for de novo hepatic lipogenesis, lipid storage, and nonalcoholic fatty liver disease pathogenesis [23,48]. Higher expression levels of *FABP4* and *SREBP1* mRNA in the liver were noted after chronic risperidone treatment, which resulted in liver injury due to considerable lipid infiltration in the liver. Subsequently, serum ALT and AST levels were evaluated. Similar reports were found in mice and humans regarding long-term treatment with risperidone inducing visceral adiposity associated with cholestatic hepatitis and hepatic steatosis [18,49]. Furthermore, an evaluation of liver enzymes revealed that risperidone induced liver damage in rats through an increase in free radical damage and reduction in plasma total antioxidant activity [19]. These effects of risperidone eventually resulted in hepatic adverse effects due to the activation of hepatic lipogenesis.

We further determined that increases in the expression of FASN and PNPLA3 can cause increases in triglycerides levels, facilitate the detection of hepatic steatosis, and abolish IS [25]. Increases in circulating triglyceride levels may play a role in limiting liver lipid accumulation—the most important factor underlying nonalcoholic fatty liver disease development [22,24]; risperidone-treated obese mice exhibited elevated fatty liver scores compared with HFD-fed control mice. This effect may be related to risperidone reducing the activity of the physiological energy sensor fibroblast growth factor 21 (FGF-21; Appendix A**)**, which regulates energy metabolism by maintaining energy homeostasis through AMP-activated protein kinase/silent information regulator 1–peroxisome proliferator-activated receptor-γ coactivator-1α pathway activation. This ability implies that risperidone may be not a viable therapeutic strategy for treating nonalcoholic fatty liver disease and liver injury [25,50]. These results of decreasing energy combustion leading to obesity accord with those of other studies [24,39] and can be explained by the decrease in serum FGF-21 levels leading to a reduced ability to protect against hepatic steatosis and dyslipidemia [51] in mice with obesity. The current findings also demonstrate risperidone’s ability to increase serum ALT and AST levels (a hepatic inflammation index) in mice. This accords with the finding that lower FGF-21 levels limit the protection against fatty liver disease, which is defined as a proinflammatory condition of steatohepatitis development that is identified through histologic analysis. Risperidone induces high levels of intrahepatic lipid accumulation, thereby accelerating liver damage in diet-induced nonalcoholic fatty liver disease [52]. Moreover, FASN can activate nucleotide-binding domain leucine-rich repeat domain containing 3 (NLRP3) inflammasome-mediated caspase-1 and increase expression of *NLRP3* and levels of proinflammatory cytokines (e.g., interleukin (IL)-1β and IL-18) in obesity-induced inflammation through the inflammatory signal pathway [53]. Taken together, these findings indicate that activating multiple proinflammatory signaling pathways may be the mechanism underlying risperidone’s injurious effects on the liver. Therefore, the regulation of risperidone in these pathways could offer a novel therapeutic target for these decreasingly common sequelae of hepatic side effects after risperidone treatment.

Risperidone treatment offers protective benefits compared with medications, such as olanzapine, which can exacerbate diabetes mellitus [14] or increase insulin secretion patients with schizophrenia who do not have diabetes but exhibit signs of hypoglycemia [12,54]. Moreover, paliperidone, an analog of risperidone, causes significant reductions in glucose levels by inducing insulin release through the inhibition of 𝛼2-adrenoceptors in patients who have T2D [12]. Thus, long-term administration of risperidone for the treatment of hyperglycemia has a prominent antihyperglycemic effect. However, our study demonstrates that risperidone elevates fasting glucose levels and dramatically worsens glucose tolerance, as indicated by estimations made using the intraperitoneal glucose tolerance test (IPGTT) and the AUC for plasma levels of glucose at 120 min in a diet-induced obesity and T2D mouse model. Similarly, other reports have indicated that risperidone treatment exacerbates hyperglycemia. In contrast to other reports [12,54], our results indicate that risperidone reduced serum insulin levels, as evidence by the long-term use of risperidone resulting in a reduced β-cell percentage in pancreas sections of the mice (Appendix A). However, one study reported that risperidone treatment on its own led to no changes in blood glucose level in normal rats and rabbits [55]. The fact that relevant data in rodents did not match human data is likely attributable to the side effects that antipsychotics have on animals; examples are stiffening of the muscles and sedation, which lead to reduced activity and metabolic alterations. These effects consequently interfere with the influences of antipsychotics on metabolism, satiety, and dietary intake in obese mice, particularly in the short term [25]. However, research on animal models has provided valuable data in terms of the mechanisms underlying the antipsychotics’ ability to induce or mitigate increases in weight, but findings have differed for different strains, species, medication administration approaches, and how animals were handled and maintained. The observation of diabetes signs following the continuation of risperidone treatment led to our belief in a possible association between glucose intolerance due to risperidone and glucose metabolism in obesity development. Thus, medical personnel should monitor baseline blood levels of glucose in patients with high-risk T2D and psychotic disorders who are receiving risperidone treatment, especially in terms of monitoring for temporal hypoglycemia, which can make diabetes difficult to recognize.

As substantial risk factors for inflammation, obesity and increases in weight are pathogenically involved in IR and T2D development and progression [56]. Moreover, in the present study, the mice receiving risperidone treatment exhibited elevations in IR index and declines in IS index, which are prominent pathophysiological indicators of IR and insulin action [25]. The high IR index in the mice on an HFD receiving risperidone treatment can be explained by their increases in body weight gain, fatty liver scores, and fat pad weights; for these factors, the control mice displayed decreases. In accordance with the findings of other studies [57,58], oxidative stress was also found to play a significant role in β-cell damage and complications in T2D. To further confirm the symptoms of diabetes and IR in risperidone-treated obese mice, we analyzed their pancreatic antioxidant enzyme activity. Research has indicated that diabetes is associated with decreased antioxidant enzyme activity [59], and we observed reductions in catalase, GPx, and SOD activities in the pancreases of the risperidone-treated HFD-fed mice compared with the control mice (Appendix A). The data suggest that risperidone treatment attenuates pancreatic antioxidant enzyme activity, thereby facilitating to develop IR. A subacute, chronic inflammatory state is often the result of accumulating excess lipids in adipose tissue and the liver and may be a risk factor for IR, as evidenced by changes in biochemical markers of inflammation [32,56]. Indeed, higher serum cytokines levels (e.g., those of IL-1β and tumor necrosis factor-β (TNF-β)) were observed in risperidone-treated mice receiving an HFD compared with the control mice (Appendix A). Cytokine dysregulation can accelerate the development of obesity-related IR and subsequently impair glucose homeostasis and IS. However, risperidone reduces IS index, which can be explained by lower insulin levels in risperidone-treated HFD-fed mice compared with control mice. Our results also indicate the possibility that risperidone impairs glucose homeostasis and insulin signaling by reducing muscle Akt phosphorylation, serum FGF-21 levels (Appendix A), and GLUT4 expression and regulates metabolism by altering glucose flux in the liver and improving the IS in obese mice [25]. Notably, GLUT4 is key in the regulation of glucose homeostasis and insulin action, and decreased GLUT4 expression leads to decreases insulin-mediated glucose uptake [32,36], which could be due to exacerbated hyperglycemia, evidenced by increased glucose intolerance according to the IPGTT. The aforementioned results indicate the influence risperidone has on IS through the exacerbation of hyperglycemia induced by a long-term HFD because decreased insulin signaling exacerbates problems with metabolism, resulting in the inability to maintain glucose homeostasis.

One noted side effect of the long-term administration of psychotropic drugs is kidney damage, which is attributable to the association of these drugs with adverse effects on obesity, metabolic syndrome, or diabetes [60,61]. Regarding streptozotocin-induced diabetic nephropathy in rats, hyperglycemia causes injury to renal glomeruli, vessels, and renal tubules, which may cause diabetic nephropathy, associated with kidney inflammation and further promote CKD [62]. We found that risperidone-treated mice exhibited glomerulonephritis as an adverse effect on their kidneys, which is in line with reports that have suggested that various psychotropic drugs (such as aripiprazole, chlorpromazine, clozapine, olanzapine, ziprasidone, and sertindole) cause renal injury on renal dysfunction, including glomerulonephritis, interstitial nephritis, and CKD [61,63]. Moreover, interstitial nephritis is characterized by inflamed renal tubulointerstitium [21]. One patient presented with symptoms of clinical nephrotoxicity, namely elevation of blood creatinine and BUN levels [63]. Taken together, long-term administration of risperidone can be considered to cause renal damage with inflammatory cell infiltration in the renal interstitium in tandem with increases in major markers of nephropathy (creatinine and BUN levels). Kidney function should be carefully monitored in patients receiving risperidone to ensure timely identification of this rare side effect.

Similar to other works, we found that risperidone administration induced renal damage [20,63,64]. Under hyperglycemic conditions induced to simulate the risk factors for risperidone, risperidone-treated mice exhibited a reduction of antioxidative defense enzymes in the kidneys. Lower antioxidative enzyme levels limit the protection against free radical damage, which could lead to renal inflammation in rats [65,66]. Immunohistochemical (IHC) staining revealed that risperidone-treated mice had higher levels of renal inflammatory cytokines, such as IL-1β (Appendix A). However, a clinical survey reported that risperidone potentially exerts antioxidative effects through a reduction in lipid peroxidation and increase in antioxidant protection from lipid peroxidation related to paraoxonase 1 [67]. Unlike our risperidone-treated mice with hyperglycemia, the samples in the aforementioned clinical survey were emergency cases and had untreated psychosis for 61 days (median). The discrepancies of our results with those of other studies are probably indicative of the effects of staging for hyperglycemia. Therefore, in the beginning stages of metabolic syndrome, hyperglycemia could be defined as reduced paraoxonase 1 activity and corresponding increases in enzymatic antioxidants, which generally mitigate relative elevations in ROS and lipid peroxidation. However, long-term continuous risperidone use can lead to hyperglycemia-induced kidney injury along with chronic inflammation, thus adding to injury caused by oxidative stress and reductions in antioxidant levels. Thus, a specific renal damage profile and the side effect of hyperglycemia rather than changes in antioxidant parameters seem to have an association with anti-psychotropic medications in patients with psychotic disorders. A greater follow-up duration may clarify risperidone’s long-term impacts.

A HFD was successful for establishing an animal model of obese and hyperglycemic mice, and risperidone effectively accelerated their weight gain and increased their food intake, adipocyte size, fat pad weight, fatty liver scores, and serum triglyceride, ALT, and AST levels. This finding could be associated with accumulating lipids and activating lipogenesis for the regulation of nutrient metabolism. In the mice, the administration of risperidone exacerbated hyperglycemia and reduced glucose tolerance with interstitial nephritis, and decreased GLUT4 expression in the skeletal muscles may have played a role in the aforementioned glucose metabolic changes. These findings indicate deterioration in glucose homeostasis and IS. Moreover, in the mice, an HFD accelerated the deterioration of glucose homeostasis, and risperidone accelerated diabetes symptom development. This suggests that risperidone could have side effects related to diabetes, fatty liver disease, and renal damage and that taking it in the long term could lead to exacerbated diabetes, obesity control, and CKD.

## 4. Materials and Methods

### 4.1. Animals, Diet-Based Obesity Induction, and Risperidone Therapy

We procured C57BL/6J mice (sex: male; age: 5 weeks) from Education Research Resource, National Laboratory Animal Center, Taiwan. In accordance with Taiwan government’s recommendations, animal housing and experimentation were performed following the Guidelines for the Care and Use of Laboratory Animals. The review of our experimental protocol was conducted by National Chiayi University’s Institutional Animal Care and Use Committee, who approved it under the approval No. 109019.

The mice were continually fed an HFD diet (diet 592Z, containing 20.4% of protein and modified laboratory with 35.5% lard, with 4.5 kcal/g metabolizable energy; PMI Nutrition International, Brentwood, MO, USA) for 10 weeks. To establish obesity, our mice were administered the HFD for 10 weeks longer than the typical duration applied in other studies (i.e., 4 weeks) [25]. The groups were subsequently separated to form two subgroups: one receiving 1 mg/kg oral risperidone (Sigma-Aldrich, St. Louis, MO, USA) and the other (control group) receiving saline through daily gavage for the final 56 days of the diet period (36.71 ± 0.69 g in risperidone-treated mice and 36.69 ± 0.61 g in control mice (vehicle treated), *p* > 0.05). Literature on risperidone as a potential drug for application, in studies on behavior, antipsychotic disease, immunosuppression, and cardiovascular disease in a mouse model was referenced when selecting the risperidone dose [68,69,70]. During the study, mice were separately kept in microisolation cages placed on racks ventilated by air filtered by high-efficiency particulate air filters under temperature and humidity controlled at 22 ± 1 °C and 55% ± 5%, respectively, under a 12:12-h light/dark cycle, all with free water and food access. We monitored body weight and food intake on a weekly basis from experiment initiation.

At the end of the experiment, we euthanized all mice and harvested their blood and various tissues for further analysis. Furthermore, we evaluated the impacts of orally administered risperidone on the weight, food intake, adipocyte content, biochemical changes, blood glucose level, fatty liver scores, endocrine profiles, hepatic lipogenesis, insulin signaling expression, and renal pathology of the mice.

### 4.2. Food Intake, Body Weight, Leptin, and Insulin Level Measurement

We measured food intake and body weight at each week during the study period. The weight of food remaining in individual cage dispensers and on the floor was determined to measure food intake. Enzyme-linked immunosorbent assay kits (#90030 and #INSKR020; Crystal Chem, Downers Grove, IL, USA) were used for measuring serum leptin and insulin levels in harvested blood and tissues.

### 4.3. Measurement of Serum and Hepatic Levels of Triglycerides and Serum Levels of Creatinine, Bun, ALT, and Ast

Blood samples were used to measure serum levels of triglycerides, ALT, AST, creatinine, and BUN on Catalyst One Chemistry Analyzer (IDEXX Laboratories, Westbrook, ME, USA) and commercial kits, all in accordance with the manufacturer’s instructions. All coefficients of variation between and within analysis runs were <2%. Following a related study [71], hepatic triglycerides were extracted from homogenized liver samples using a Triton X-100 solution. The extracted samples were then solubilized through two cycles of gradual heating to 90 °C over a period of 5 min and subsequent cooling to room temperature; the solution was then centrifuged to eliminate insoluble material. The supernatant was finally collected for colorimetric assay–based triglyceride analysis by using the colorimetric assay kit Triglyceride Quantification Kit from BioVision (Milpitas, CA, USA).

### 4.4. IPGTT

Following 49-day treatment with risperidone or saline, the IPGTT was conducted (dose = 1 g·of glucose/kg of body weight) on mice after subjection to overnight fasting with ad libitum access to water. At 0, 30, 60, 90, and 120 min after intraperitoneal injection of glucose, blood glucose levels were measured from blood extracted from the tails of the mice using a One Touch glucose meter (LifeScan, Milpitas, CA, USA). We assessed glucose tolerance by calculating the glucose tolerance AUC (0–120 min).

### 4.5. IR and IS Indices

Fasting glucose has been widely used in research to estimate HOMA-IR index and IS index [24,25,32,35]. Therefore, these two indices were employed for evaluation of mouse IR and insulin function following risperidone treatment. The calculation of HOMA-IR and IS indices [25] was performed using the following formulas:-HOMA-IR index = [fasting insulin (in mU/L) × fasting glucose (in mmol/L)]/22.5,-IS index = [1/fasting insulin (in mU/L) × fasting glucose (in mmol/L)] × 1000.

These calculations were based on fasting plasma levels of glucose and insulin in accordance with the HOMA method, which has been validated against clamp measurements.

### 4.6. Histological and Morphometric Analyses of Tissues

The weights of the liver, heart, kidney, spleen, EWAT, and RWAT were measured, and these weights were determined as total body weight percentage. We determined hepatic fat infiltration through H&E staining and scored from 0, 1, 2, 3, and 4 for 0%, <5%, 5%–25%, 25%–50%, and >50% fat infiltration of the liver surface, respectively [25,32]. Numerous retroperitoneal and epididymal adipose tissue sections were obtained and underwent systematic analysis to determine adipocyte sizes. Fad sections underwent H&E staining. A minimum of 10 fields (reflecting roughly 100 adipocytes) on each slide underwent analysis for each sample [24,25,32]. Next, tissues obtained from bisecting the kidneys along the longitudinal axis were prepared and subjected to H&E staining. The origin of the specimens was concealed for the detection of interstitial nephritis in accordance with a previously established method [72]. We employed Moticam 2300 (Motic Instruments, Canada), a high-resolution digital microscope equipped with Motic Images Plus (version 2.0) was used to capture images and analyze adipocyte size distributions between control and treated obese mice.

### 4.7. RNA Extraction and Real-Time Quantitative Polymerase Chain Reaction

We employed TRI Reagent (Sigma-Aldrich) to extract total RNA in liver tissues and assessed RNA concentration on the basis of absorbance at 260–280 and 230–260 nm on a Qubit fluorometer (Invitrogen, Carlsbad, CA, USA). Next, we reverse transcribed the RNA (1 μg) into cDNA by using iScript cDNA synthesis kit (Bio-Rad, Hercules, CA, USA)—in accordance with the manufacturer instructions. We then performed real-time polymerase chain reaction (PCR) using this cDNA and iTaq universal SYBR Green Supermix (Bio-Rad)—all in accordance with the manufacturer protocol. *F**ABP4* and *SREBP1* mRNA expression levels were specifically determined using CFX Connect Real-Time PCR System (Bio-Rad). The PCR was performed as follows: 95 °C for 5 min, and then 45 cycles at 95 °C for 15 s, followed by 60 °C for 25 s. The *FABP4* sequence primers employed in this study were forward, 5′-GATGAAATCACCGCAGACGACA-3′ and reverse, 5′-ATTGTGGTCGACTTTCCATCCC-3′ [73]; those for *SREBP1* were forward, 5′-CGG AAGCTGTCGGGGTAG-3′ and reverse, 5′-GTTGTTGATGAGCTGGAGCA-3′ [74]. Each target gene expression level was calculated relative to the Actb levels and expressed using the 2^−ΔΔCt^ method.

### 4.8. Western Blotting

After each experiment, the mice were euthanized using an overdose of anesthetic combined with carbon dioxide. The liver and gastrocnemius muscles of the mice were obtained quickly, coarsely minced, and homogenized. We performed western blotting as described elsewhere [32] and used antibodies against PNPLA3 (Sigma-Aldrich) as well as those against FASN, GLUT4, actin, phospho-Akt (Ser473), and Akt (Cell Signaling Technology, Beverly, MA, USA). We used enhanced chemiluminescence reagents (Thermo Scientific, Rockford, MA, USA) to produce immunoreactive signals and UVP ChemStudio (Analytik Jena, Upland, CA, USA) to detect these signals. Protein expression and phosphorylation were quantified using ImageJ by National Institutes of Health (Bethesda, MA, USA).

### 4.9. Measurement of Renal Catalase, GPx, and SOD Levels

To measure renal catalase, GPx, and SOD activities, we prepared homogenized kidney lysates using 0.1 M Tris/HCl (pH 7.4) containing 0.1 mg/mL phenylmethanesulfonyl fluoride, 5 mM β-mercaptoethanol, and 0.5% Triton X-100 and removed fresh renal cortical tissue, as previously described [75]. Subsequently, the supernatant was obtained for analysis after centrifugation at 14,000× *g* for 5 min at 4 °C. Next, catalase GPx, and SOD levels in the kidney samples were measured using a commercially available colorimetrical kit (#K773-100 for catalase, #K762-100 for GPx, and #K335-100 for SOD; BioVision, Milpitas, CA, USA) in accordance with the manufacturer instructions.

### 4.10. Statistical Analysis

All results are presented as means ± standard deviation. The *t* tests were used to analyze differences in the two mouse groups. Analysis of variance and subsequent post hoc Bonferroni testing were applied for the determination of differences in analyses of three or more groups. Significance was set at *p* < 0.05. We employed Fisher exact test for evaluation of the significance in contingency data.

## 5. Conclusions

Taken together, the results of this study highlight a crucial potential side effect of risperidone in obese mice. Our findings provide novel evidence that the continuous administration of risperidone in mice exacerbates obesity and hyperglycemia and causes renal damage in response to an HFD. This observation is associated with the deterioration of metabolic homeostasis through both increases in fatty liver scores and decreases in glucose transporter expression. Our study reveals that risperidone treatment in mice causes weekly body weight gain and changes in body weight, fat tissue weight, fatty liver disease, food efficiency, ALT and AST serum levels, serum and hepatic triglyceride levels, adipocyte size, and renal pathology. The deterioration in regulation of adipose lipogenesis and accumulation of lipids in the liver may contribute to an individual’s susceptibility to developing obesity-related metabolic disturbances. The data presented in this study suggest that risperidone treatment reduces pancreatic antioxidant enzyme activity, increases serum inflammatory cytokines, and causes β-cell damage. Moreover, we present several lines of evidence to support that risperidone exacerbates hyperglycemia and that its reduction in GLUT4 expression and Akt phosphorylation in insulin signaling is associated with impaired glucose tolerance, increased IR, and decreased IS. Nephropathy in risperidone-treated obese mice may be due to exacerbated hyperglycemia and reductions in antioxidant enzyme activity. Thus, the distinct acceleration in obesity development, worsened hyperglycemia, and greater renal injury response observed in HFD-fed mice receiving risperidone treatment supports the notion that risperidone is a psychotropic agent that may increase an individual’s likelihood of developing related metabolic abnormalities. In addition to blood glucose changes, hepatic and renal function should be carefully monitored after the initiation of risperidone to achieve early recognition of this rare side effect, particularly in patients with schizophrenia, which would enable prompt identification and treatment for the prevention of cachexia, obesity, and hyperglycemia.

## Figures and Tables

**Figure 1 ijms-22-00409-f001:**
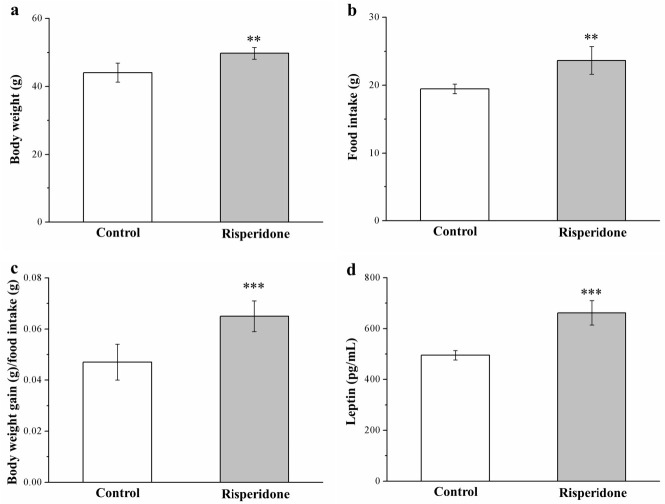
Changes in (**a**) body weight, (**b**) food intake (per mouse per week), (**c**) daily food efficiency, and (**d**) serum leptin levels in the 1 mg/kg/day risperidone-treated mice and the control group over the period of 56 days. All data for both groups are presented as means ± standard deviation (*n* = 10). ** *p* < 0.01, *** *p* < 0.001.

**Figure 2 ijms-22-00409-f002:**
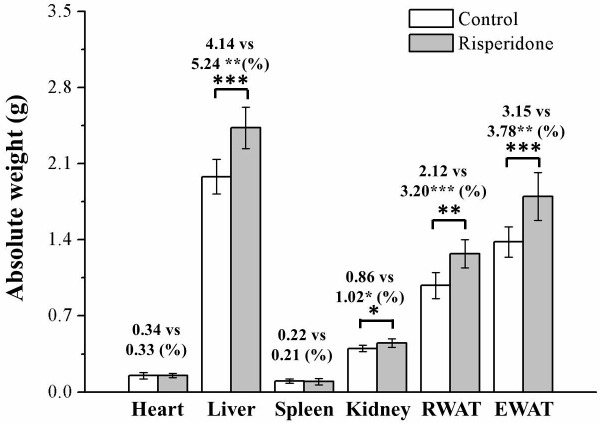
White adipose tissue and organ absolute weights, and weights of all tissues and organs, after normalization for body weight (%) in the 1 mg/kg/day risperidone treatment and control mice over the period of 56 days. All data for both groups are presented as means ± standard deviation (*n* = 10). * *p* < 0.05, ** *p* < 0.01, *** *p* < 0.001. EWAT, epididymal white adipose tissue; RWAT, retroperitoneal white adipose tissue.

**Figure 3 ijms-22-00409-f003:**
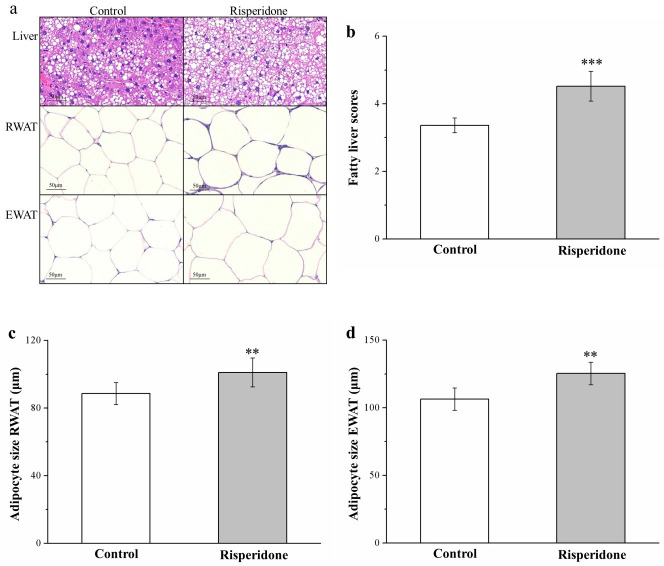
(**a**) Hematoxylin and eosin (H&E)-stained sections of control and treated mouse liver, EWAT, and RWAT (magnification, 200×). (**b**) Fatty liver score changes. (**c**,**d**) Changes in RWAT (**c**) and EWAT (**d**) adipocyte sizes in mice receiving 1 mg/kg/day risperidone treatment and the control group over the course of 56 days. All data for both groups are presented as means ± standard deviation (*n* = 10). ** *p* < 0.01, *** *p* < 0.001.

**Figure 4 ijms-22-00409-f004:**
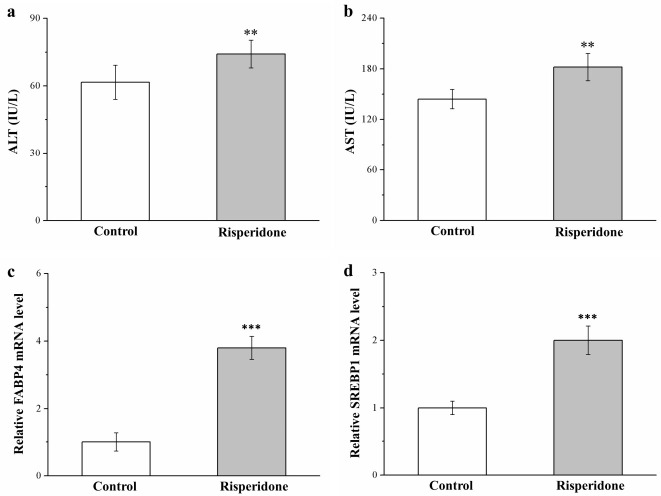
Changes in serum (**a**) alanine aminotransferase (ALT) and (**b**) aspartate aminotransferase (AST) levels and in (**c**) *FABP4* and (**d**) *SREBP1* mRNA levels in the livers of treated and control mice over the course of 56 days. All data for both groups are presented as means ± standard deviation (*n* = 10). ** *p* < 0.01, *** *p* < 0.001.

**Figure 5 ijms-22-00409-f005:**
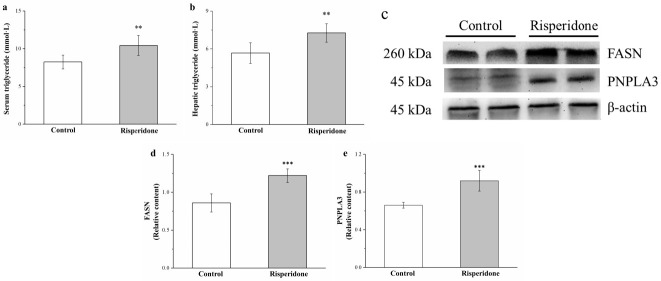
Changes in serum and hepatic triglyceride level and fatty acid synthase (FASN) and patatin-like phospholipid domain containing protein 3 (PNPLA3) expression. (**c**) A representative Western blot of liver extracts for FASN and PNPLA3 expression. Changes in (**a**) serum and (**b**) hepatic triglyceride levels as well as (**d**) FASN and (**e**) PNPLA3 expression in the control and treated mice over the course of 56 days. All data for both groups are presented as means ± standard deviation (*n* = 10). ** *p* < 0.01, *** *p* < 0.001.

**Figure 6 ijms-22-00409-f006:**
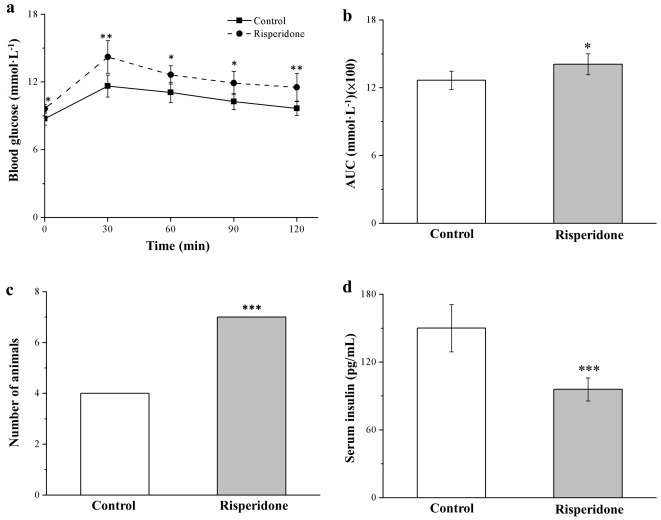
Changes in (**a**) intraperitoneal glucose tolerance (1 g of glucose/kg body weight), (**b**) under the curve (AUC) at 120 min after glucose injection, (**c**) glucose intolerance (Fisher’s exact test), and (**d**) serum insulin levels in the risperidone treatment (1 mg/kg/day) and control group over the 56-day treatment period. All data for both groups are presented as means ± standard deviation (*n* = 10). * *p* < 0.05, ** *p* < 0.01, *** *p* < 0.001.

**Figure 7 ijms-22-00409-f007:**
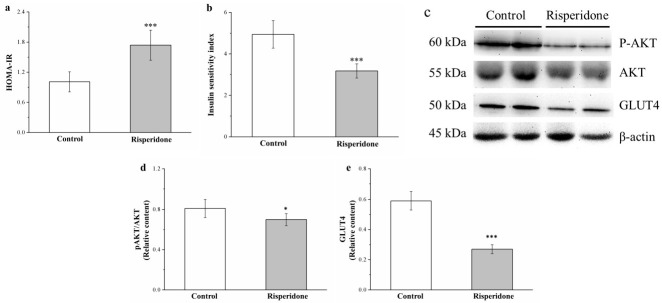
Changes in (**a**) homeostatic model assessment for insulin resistance (HOMA-IR) and (**b**) insulin sensitivity (IS) indices, (**c**) a representative blot of muscle extracts, (**d**) Akt phosphorylation, and (**e**) glucose transporter 4 (GLUT4) expression in the gastrocnemius muscle in the control and 1 mg/kg/day risperidone-treated mice over the period of 56 days. All data for both groups are presented as means ± standard deviation (*n* = 10). * *p* < 0.05, *** *p* < 0.001.

**Figure 8 ijms-22-00409-f008:**
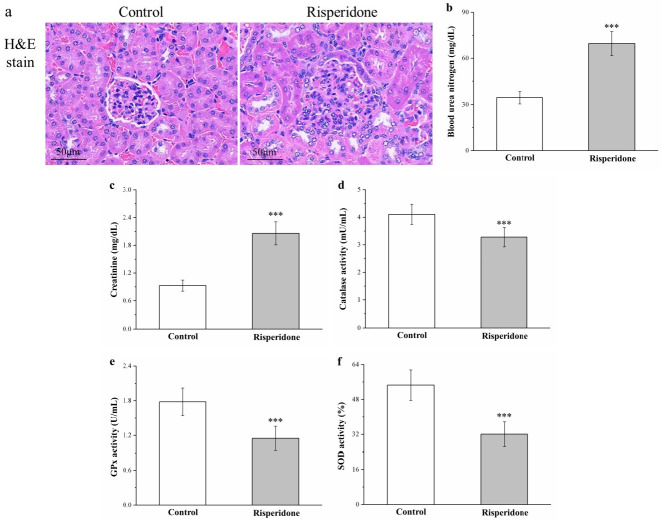
Changes in (**a**) renal morphology according to H&E staining (magnification, 200×), (**b**) serum blood urea nitrogen (BUN) levels, (**c**) serum creatinine levels, (**d**) renal catalase activity, (**e**) renal glutathione peroxidase (GPx) activity, and (**f**) renal superoxide dismutase (SOD) activity in the 1 mg/kg/day risperidone treatment and control groups over the 56-day treatment period. All data for both groups are presented as means ± standard deviation (*n* = 10). *** *p* < 0.001.

**Table 1 ijms-22-00409-t001:** Effects of risperidone on the size distribution of fat cells in high-fat diet (HFD)-fed mice with and without 1 mg/kg/day risperidone treatment.

Variable	Control	Risperidone
RWAT		
Adipocyte diameter		
0–50 μm (%)	21.67 ± 2.90	12.67 ± 2.88 ***
50–100 μm (%)	48.33 ± 3.85	27.67 ± 3.57 ***
100–150 μm (%)	28.33 ± 2.18	42.99 ± 3.22 ***
>150 μm (%)	1.67 ± 0.25	16.67 ± 3.05 ***
EWAT		
Adipocyte diameter		
0–50 μm (%)	8.33 ± 1.48	0 ± 0 ***
50–100 μm (%)	30.01 ± 2.97	3.33 ± 0.37 ***
100–150 μm (%)	33.33 ± 3.25	40.00 ± 3.57 **
>150 μm (%)	28.33 ± 2.59	56.67 ± 4.94 ***

All data for both groups are presented as means ± standard deviation. *** *p* < 0.001. *n* = 10 for both groups.

## Data Availability

The data presented in this study are available on request from the corresponding author.

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
