# Peer review of "Risperidone Exacerbates Glucose Intolerance, Nonalcoholic Fatty Liver Disease, and Renal Impairment in Obese Mice"

_ijms, 2021, doi:10.3390/ijms22010409_

Round 1

Reviewer 1 Report

Dear, Lan-Szu Chou, Ph.D., Li-Wei Hsiao, Ph.D. and Geng-Ruei Chang, Ph.D.

-------------------------------------------------------------------------------------------------------------------------

O.K.

You have revised.

-------------------------------------------------------------------------------------------------------------------------

Author Response

Response:

We appreciate the reviewer’s comments. Thank you for your suggestions for revisions again.

Reviewer 2 Report

  1. Authors should control the appropriateness of references, i.e., Nonalcoholic fatty liver disease has also gained increasing recognition as a possible complication of obesity [35]?

Author Response

Response:

We appreciate the reviewer’s comment.

Revision:

To avoid unnecessarily verbose descriptions, we have deleted this sentence in the revision.

Thank you for your suggestion for revisions again. We have carefully considered your comment and have made appropriate changes to the manuscript.

This manuscript is a resubmission of an earlier submission. The following is a list of the peer review reports and author responses from that submission.

Round 1

Reviewer 1 Report

Dear, Dr. Lan-Szu Chou, Dr. Li-Wei Hsiao and Dr. Geng-Ruei Chang

Please check and revise.

-------------------------------------------------------------------------------------------------------------------------

I think this manuscript is so long. This should be revised to more smaller volume.

For example, graphs forms are changed to more thin.

-------------------------------------------------------------------------------------------------------------------------

Reviewer 2 Report

While the study is very interesting from a clinical perspective, the data presented does not strongly support the conclusions drawn by the authors.
A more detailed mechanism must be needed to explain.
For example,
Longitudinal data on insulin resistance and insulin secretion
Longitudinal data on leptin resistance,
Analysis of kidney and liver fibrosis,
Analysis of oxidative stress in the pancreas.

Reviewer 3 Report

The authors implanted a study to ascertain the effect of this drug on some metabolisms/organs of mice fed a high-fat diet (HD), but this reviewer has some criticism about the selection of two groups, one with HD and another with HD plus drug.

The idea to add a drug after an HD leaves some doubts on the possible effect of diet on hepatic metabolism of the drug, thus the modified metabolism or organ damages are due not necessarily to the drug but its bioavailability that can be modified by diet.

To ascertain the effects and related mechanisms of the drug, I have implanted a different study, i.e., four groups, one group of mice with a chow diet, the second group of mice with a chow diet and treated with the drug, the third group with HD without drug and the last group with HD and drug.

Then, data should have analyzed by ANOVA in a study 2X2 (HD yes/not, drug yes/not) to better evidence the intra- and inter-groups difference and thus draw strong conclusions.

Authors should present their data as means plus/minus SD and not SEM because readers are interested in knowing the dispersion of data and not the precision of the mean due to the paucity of observations for each group.

Authors should lessen the potential translational approach because the animal models only partially mirror human diseases.

Reviewer 4 Report

The paper “Risperidone exacerbates glucose intolerance, nonalcoholic fatty liver disease, and renal impairment in obese mice” is well written and divided into sections and subsections. In the article there are no grammatical and stylistic errors. It is sufficiently illustrated with figures and tables.

In my opinion, this review paper can be recommended for publication after minor revision.

It is recommended to add articles of 2019-2020 to the list of references.
